# Serum Analytes of American Mink (Neovison Vison) Challenged with Aleutian Mink Disease Virus

**DOI:** 10.3390/ani12202725

**Published:** 2022-10-11

**Authors:** A. Hossain Farid, Priyanka P. Rupasinghe

**Affiliations:** Department of Animal Science and Aquaculture, Faculty of Agriculture, Dalhousie University, Truro, NS B2N 5E3, Canada

**Keywords:** American mink, Aleutian mink disease virus, Box–Cox transformation, serum analytes, sex, tolerance

## Abstract

**Simple Summary:**

Aleutian mink disease virus (AMDV) causes major health problems in the mink industry worldwide. The disease caused by AMDV has no cure or effective vaccine, and long-term viral eradication programs have failed in many countries. Some AMDV infected mink are genetically capable of tolerating the infection and living healthy and productive lives. Genetic selection for tolerance is, thus, a practical strategy to combat this virus. Accurate identification of tolerant animals is the fundamental issue in selection programs. The concentrations of some blood analytes, which are widely used as indicators of the presence and severity of diseases in humans and animals, are known to increase the accuracy of identifying tolerant mink. The objective of this study was to evaluate the merits of 14 serum analytes as biomarkers of tolerance to AMDV infection. Blood samples from 493 AMDV inoculated mink collected between 120 and 1211 days post-inoculation were analyzed. Total serum protein and globulin were found to be the most useful biomarkers of tolerance, whereas the relationships of other serum analytes to tolerance were weak or negligible.

**Abstract:**

Black American mink (*Neovison vison*), which had been selected for tolerance to Aleutian mink disease virus (AMDV) for more than 20 years (TG100) or were from herds that have been free of AMDV (TG0), along with their progeny and crosses with 50% and 75% tolerance ancestry, were inoculated with a local isolate of AMDV. Blood samples were collected from 493 mink between 120 and 1211 days post-inoculation, and concentrations of 14 serum analytes were measured. Distributions of all analytes significantly deviated from normality, and data were analyzed after Box–Cox power transformation. Significant differences were observed among tolerant groups in the concentrations of globulin (GLO), total protein (TP), alkaline phosphatase, urea nitrogen, and calcium. Concentrations of GLO and TP linearly and significantly decreased with an increasing percentage of tolerance ancestry. Eleven analytes had the smallest values in the tolerant groups (TG100 or TG75), and eight analytes had the greatest values in the non-selected groups (TG0 or TG50). Antibody titer had the greatest correlation coefficients with GLO (0.62), TP (0.53), and creatinine (0.36). It was concluded that selection for tolerance decreased the concentrations of most serum analytes, and TP and GLO were the most accurate biomarkers of tolerance to AMDV infection. Males had significantly greater values than females for phosphorus and total bilirubin concentrations, but females had significantly greater amylase, cholesterol, and BUN concentrations than males.

## 1. Introduction

Aleutian mink disease virus (AMDV) is the causative agent of Aleutian disease (AD), which decreases productivity and increases mortality in mink [1]. The disease has no cure or effective vaccine [2], and extensive virus eradication programs have failed in many countries [3,4,5]. Some AMDV infected mink exhibit mild sub-clinical symptoms and live healthy and productive lives [1,6,7,8], a phenomenon which is genetically controlled [9,10], suggesting that genetic selection for tolerance is a practical strategy to combat this virus. The most efficient method for identifying mink tolerant to AMDV infection remains a matter of debate. Tolerant herds of mink have been established over the course of more than 20 years by farmers using iodine agglutination tests (IATs) [6,11]. The contribution of IATs in response to selection appears to have been insignificant [6], because gamma globulin is elevated in response to infection by all pathogens [12] and IAT results for individual mink greatly fluctuate over time [6,13,14].

Infected mink that remained healthy (non-progressive infection) have long been known to have low antibody titers [15,16,17]. Antibody titer is also negatively associated with reproductive performance [18,19], thus, implying that selection for low antibody titers is a logical approach for establishing tolerant herds. Development of enzyme-linked immunosorbent assay (ELISA) systems [20,21,22] has facilitated antibody titer measurement for the identification of tolerant mink, and a combination of weak ELISA positive and normal litter size was effective in reducing the incidence of clinical forms of the disease [8]. In addition to their pathogenic roles, antibodies play protective roles, because they restrict AMDV replication [23,24], and support partial viral clearance and sequestration of the virus during the early phases of infection [24,25]. Some seropositive mink have been found to remain healthy for more than 500 dpi [15,26], and to live longer than seronegative mink monitored for more than 1200 dpi [14]. Furthermore, the time of seroconversion of individual mink is highly variable after natural exposure to the virus [7,27], and is influenced by viral virulence [26] and dose [26,28,29]. The occurrence of seroconversion also varies over time [7,14,15,25,29,30]. The relationship between antibody titer and the degree of tolerance is, thus, complex, and hinders the accuracy of identifying tolerant mink in naturally infected herds for which the strain and amounts of circulating viruses, and the time of establishment of infection, are unknown.

An alternative approach to antibody titer is the use of viremia, an indicator of viral replication. The viral concentration in the blood has a pronounced effect on the severity of AD lesions in the kidneys [25]. Viremia has negative effects on animal health and productivity [7,17], and non-progressively infected mink show transient viremia [17]. As with antibody titer, viremia is influenced by several factors that are often difficult to control, including the time after infection [7,17,23,25] and the viral dose [28]. Selecting tolerant mink solely on the basis of antibody titer or viremia is, thus, subject to inaccuracy, and the inclusion of health and productivity information in the selection criteria would increase precision when identifying tolerant mink.

The incorporation of information regarding the degree of damage to the organs known to be most severely affected by AD in the evaluation procedure could also improve the accuracy of the estimation of tolerance. Changes in the concentrations of blood analytes are useful indicators of tissue damage in many diseases and are routinely used as diagnostic tools to assess human and animal health status [31,32]. Concentrations of analytes are elevated in the blood because of their leakage from damaged cells, and the rate of leakage is influenced by the concentration of the substances in the cells, their rate of clearance from plasma, or potentially elevated cell membrane permeability in the organs where these substances are synthesized [31,32]. The effects of AMDV infection on blood analytes of mink is limited to one report in which 49 AMDV infected and 25 healthy pastel and Aleutian mink of both sexes were compared. The infected animals had significantly higher concentrations of serum urea nitrogen (BUN), total protein (TP), globulin (GLO), and amylase (AMYL); they also showed significantly lower albumin (ALB), albumin/globulin ratio (A/G), and calcium (Ca), as well as non-significantly higher inorganic phosphorus (PHOS), cholesterol (CHOL), and alkaline phosphatase (ALKP); there was no effect on blood uric acid, creatinine (CREA), glucose (GLU), and total bilirubin (TBL) [33]. More information is needed before data of blood analytes are incorporated into the procedure of identification of tolerant mink.

Distributions of blood analytes often deviate from normality [14,34,35], and the assumption of homogeneous variances may also be violated; thus, data need to be transformed to meet the conditions for the parametric statistical analysis [36]. The lack of normality may also negatively influence estimates of fixed effects [37]. The details of statistical analyses of blood analytes, including normality testing, and transformation and back-transformation of the means and standard errors, are often unclear in published reports. For instance, no indication of the normality testing prior to a multi-way analysis of variance for blood analytes was reported in mink [38,39,40,41], dogs [42], cats [43], and chickens [44]. Blood analytes have also been compared between two [33,45,46,47,48] or more groups [49,50,51,52,53,54,55] with a t-test without prior normality testing. The logarithmic transformation has normalizing and variance equalizing effects on positively skewed distributions [36,56,57,58], and has frequently been applied to some blood analytes to achieve normality [42,59] or homogeneity of variances [45,60]. However, log-transformed data may not become normally distributed [61], and transformed data have not subsequently been tested for normality or homogeneity of variances.

The means of transformed data need to be back-transformed to the original scale to be meaningful [36,58]. The means of log-transformed data are not the log of the means, and the antilogarithm of the means on log scale does not give an estimate of the means on the original scale; instead, they correspond to the geometric means on the original scale, which are always less than or equal to the arithmetic means [36,56,58]. In several previous studies, the data have been log-transformed before parametric statistical analysis, and the means and standard errors have been reported after back-transformation to the original scale without explanation of the method of back-transformation [34,45,57,60], and the values have only occasionally been referred to as the geometric means [59].

Although log-transformation may be helpful for decreasing the degrees of deviation of distributions of blood analytes from normality, it may not be the optimal method for all analytes because of the considerable differences in the shapes of their distributions. The Box–Cox family of power transformations [62] may provide an appropriate method of transformation to achieve normality and homogeneity of error variances for individual analytes. The transformation would be specific for each analyte because the transformed scale is a function of the distribution of each analyte, and has several advantageous over log transformation [63]. Although Box–Cox transformation does not ensure the normality of the transformed data, it would decrease the degree of deviations of distributions from normality, and is expected to reduce problems with estimation, prediction, and inference [64], including more precise estimates of fixed effects after analysis of variance [37]. The major challenge in using Box–Cox transformation is the bias associated with the back-transformed estimates in the original scale, thus, potentially explaining the reason that this method has rarely been used in the analysis of blood parameters in animal research. Several adjustment procedures for decreasing the transformation bias have been suggested [61,65,66], but these are not easy to implement.

The objective of this study was to assess the effects of previous selection for tolerance on serum profiles of mink challenged with AMDV. The information may provide a useful tool for the detection of animals that can tolerate AMDV infection. Furthermore, the analysis of 14 serum analytes with different distributions on large numbers of mink enabled evaluation of the effects of Box–Cox power transformation on the results of the analysis of variance.

## 2. Materials and Methods

### 2.1. Sources of Animals and Animal Management

Black American mink (*Neovison vison*) from a farm that have been selected for tolerance to AMDV for more than 20 years, by using IAT [6], are referred to as the tolerant group (TG100), and mink from farms that have been free of AMDV for several years (TG0), as confirmed by counter-immunoelectrophoresis (CIEP), were used. The standard CIEP test was performed on plasma at the Animal Health Laboratory of the Nova Scotia Department of Agriculture in Truro, Nova Scotia, Canada, which is accredited for this test by the Standards Council of Canada. The test was performed using a cell-cultured antigen supplied by the United Vaccine, Inc., Madison, Wisconsin, USA. Progeny of TG0 and TG100 and their crosses with 25% (TG25), 50% (TG50), and 75% (TG75) tolerance ancestry, born over the course of 4 years (2010 to 2013 inclusive), were also used in this study. Animals were kept at an enclosed bio-secure facility (Aleutian Disease Research Centre) and managed according to the standard industry practices. Animals were fed commercial dry pellets (National Feeds Inc., Maria Stern, OH, USA). Animals had free access to feed and water. Standard operating procedures were prepared for animal management and sampling according to the standards of the Canadian Council for Animal Care. Detailed information about the mink, their management practices, and the selection of replacements has been previously reported [7].

### 2.2. Source of the Virus and Inoculation Procedure

Between 2010 and 2013, a total of 1742 mink were inoculated with 10% (*w*/*v*) passage 2 of a local isolate of AMDV prepared from the spleens of inoculated mink. Animals were sedated prior to inoculation as previously described [67], and 30 µL of viral homogenate was deposited into each nostril, thus, mimicking natural infection. The amount of inoculant corresponded to approximately 300 to 700 ID_50_, depending on the method of detection (CIEP or PCR) and time of sampling (35 or 56 dpi) [28].

### 2.3. Sampling and Sample Preparation

Extra animals, and those with poor heath or low reproductive performance, were terminated during January and February from 2011 to 2014, inclusive. The experiment was terminated, and the remaining animals were killed during November to December of 2014. Animals were terminated between 120 and 1211 dpi, as previously described [67], and blood was collected into 4 mL plain vacutainers by heart puncture from anesthetized animals. Blood samples were kept in a refrigerator at the farm, then transferred in a cooler from the farm to the laboratory, which was 50 km away, and kept in a refrigerator overnight. Serum was collected by centrifugation of clotted blood at 1397 g (Porta Spin C826 centrifuge, Unico, Dayton, NJ, USA) for 10 min, then divided into 0.5 mL aliquots and stored at −80 °C until use. Additional blood samples were collected in heparinized tubes for plasma preparation for measurement of antibody titer. 

### 2.4. Laboratory Procedures

Serum samples were thawed at room temperature, and the entire content of each tube was used to measure the concentrations of ALB, GLO, TP, BUN, CREA, GLU, TBL, Ca, PHO, CHOL, and AMYL, and the activities of ALKP, alanine aminotransferase (ALT,) and gamma-glutamyl transferase (GGT) were measured with the VetTest Chemistry Analyzer (IDEXX International, http://www.idexx.dk/smallanimal/inhouse/vetlab/vettest-chemistry.html, accessed on 19 December 2019). Antibody titer at termination was measured in 11 two-fold serially diluted plasma samples (1/1 to 1/1024) with CIEP [27].

### 2.5. Data Analysis

Data were analyzed in SAS, Version 9.4 for Windows (SAS Institute, Cary, NC, USA). UNIVARIATE procedure. Data were analyzed with the GLM procedure by using a model which included the fixed effects of the tolerant groups (TG0, TG50, TG75, and TG100), inoculation dates (six dates), sex, and the covariance of age at sampling in month. Two-way interactions were not significant (sex x tolerant groups, sex x inoculation dates) or were not included in the model because of missing data. Thirteen mink in TG25 were added to TG50 before analyses. Because all blood analytes significantly deviated from normality, the Box–Cox power transformation algorithm using PROC TRANSREG in SAS with the parameter MAXITER = 0 and IDENTITY (z) for z = 0 were used to normalize the data (https://support.sas.com/documentation/onlinedoc/stat/141/transreg.pdf, accessed on 21 November 2021). The values of the power parameters (λ) obtained from the Box–Cox transformation family were used to transform the data (y_i_) to y_i_^λ^ scale by using z_i_ = (y_i_^λ^ − 1)/λ if λ ≠ 0, and z_i_ = log(y_i_) if λ = 0 [62]. The transformed data were analyzed with the same model used for the original data. Prior to transformation, zero values of AMYL and GGT were changed to 0.01. The least-squares means of the transformed data (ẑ) were back-transformed in their original scale with the equation EXP[log(λẑ + 1)/λ] if λ ≠ 0, or EXP(ẑ) if λ = 0 in Microsoft Excel, without any adjustment for the transformation bias.

## 3. Results

### 3.1. Distributions of Blood Analytes

Descriptive statistics of the blood analytes of mink at termination is shown in Table 1. The means of all analytes were greater than their medians, except for ALB, Ca, and the A/G ratio, whose means were slightly smaller than the medians. Differences between means and medians were the greatest for GGT, followed by those for ALKP, ALT, AMYL, and TBL (11.6% to 17.9% of the mean), were small for BUN, CREA, and GLO (7.4% to 8.7%), and were negligible (less than 4%) for CHOL, GLU, PHOS, and TP. The coefficient of variation (CV) was greatest for GGT (420.8%), small for ALB, Ca, and TP (less than 12%), and intermediate (17.9% to 92.8%) for the other analytes.

### 3.2. Normality and Transformation

The distributions of serum analytes were positively skewed, ranging between 0.62 (GLU) and 5.91 (GGT). However ALB, Ca, and the A/G ratio were the exceptions, and had negative and small estimates of skewness (−0.13 to −0.88) (Table 2). Estimates of kurtosis of all serum analytes were positive and ranged between 0.06 (A/G ratio) and 37.06 (GGT). Distributions of all analytes significantly deviated from normality, and estimates of the Box–Cox power parameters (λ) ranged between −2.75 (TP) and +2.75 (ALB), and were zero for BUN and CREA. The skewness and kurtosis of the transformed data were closer to zero than those of the original measurements, yet all transformed distributions, except those for CREA and GLU, significantly deviated from normality. The skewness of the transformed CREA and PHOS were zero, that of GGT was positive and large (1.37), and those of the other analytes ranged between −0.90 and 0.14 (Table 2). Estimates of kurtosis of the transformed data were small, ranging between −0.47 and 0.92, except those of TP, BUN, and Ca (1.11 to 2.58).

### 3.3. Tolerant Groups

Differences among tolerant groups in the transformed data were significant for GLO, TP, ALKP, BUN, Ca, and the A/G ratio (Table 3). No clear pattern of changes in these analytes was observed on the basis of the percentage tolerance ancestry, except for GLO and TP, which linearly and significantly decreased, and the A/G ratio which linearly and significantly increased with increasing percentage tolerance ancestry. Consequently, the values of GLO and TP in TG0 were significantly greater than those in TG100, and the opposite was observed for the A/G ratio. Other significant differences were for TG75, which had a greater concentration of BUN than that in TG100, a greater concentration of Ca than that in TG50, and a smaller ALKP value than that in TG50. Other tolerant groups had intermediate values for these analytes. Although mostly non-significant, the back-transformed values of 11 analytes were smallest in the tolerant groups (TG100 or TG75) and were greatest for 8 analytes in the unselected groups (TG0 or TG50), and the A/G ratio was the only measurement with the smallest value in TG0.

Analyses of the original data showed differences with respect to the transformed data for some serum analytes (Table 4). First, the non-significant differences among tolerant groups for GGT and PHOS in the transformed data were significant in the original data. The least-squares mean of GGT in TG0 was significantly greater than those in TG50 and TG100, and the least-squares mean of PHOS in TG50 was significantly greater than those in TG75 and TG100. Second, the least-squares means of the analytes in the original data had the same rank orders as those for the back-transformed values, except for ALKP, ALT, AMYL, BUN, CREA, and GGT. The maximum and minimum values of all analytes were, however, in the same tolerant groups in both analyses, except for ALKP, AMYL, and BUN. In agreement with the results of the transformed data, the smallest least-squares means of 11 analytes were in TG100 or TG75, and the greatest least-squares means of 9 analytes were in the unselected groups TG0 or TG50.

### 3.4. Sex Effect

Analysis of the transformed data showed that males had greater values than females for nine of the serum analytes, of which the differences were significant for PHOS and TBL and approached significance for CREA (*p* = 0.08) and GLO (*p* = 0.06). The exceptions were AMYL, BUN, and CHOL, for which females had significantly greater values than males, and Ca and the A/G ratio, for which females had non-significantly greater values than males (Table 5). The results of analysis of the original data showed minor differences with respect to the transformed data. The probabilities of differences between sexes were the same in both analyses for 11 of the 14 analytes. Differences were found for AMYL and BUN, which were significant in the transformed data but became non-significant in the original data, and for ALT, which showed an opposite pattern. Except for Ca, the rank orders of the back-transformed values and least-squares means of males and females were the same in both analyses.

### 3.5. Inoculation Dates

Differences among inoculation dates were significant for all serum analytes, except BUN, Ca, and GGT, and approached significance (*p* = 0.07) for PHOS. No consistent pattern of changes in serum analytes by inoculation dates was observed (Table 6). The back-transformed values of CREA, CHOL, GLO, PHOS, TBL, and TP were the smallest for animals inoculated in 2013, i.e., closer to the termination of the experiment, and the concentrations of these analytes, except PHOS, significantly differed from the values in animals inoculated in October 2010. In contrast, animals inoculated in September 2013 had the greatest values for ALB, ALT, AMYL, Ca, and the A/G ratio among inoculation dates, and the differences were significant from those inoculated in October 2010, except for AMYL and Ca. Analysis of the original data revealed minor differences in the transformed data for some analytes (Table 7). The probabilities of differences among inoculation dates for all analytes were comparable between the transformed and original data, except for ALKP, which was significant in transformed data but non-significant in the original data. The rankings of the back-transformed values and the least-squares means were the same for ALB, Ca, CREA, CHOL, TBL, and the A/G ratio, but differences were observed in the rank orders of the values of the two analyses for ALKP, ALT, AMYL, BUN, GGT, GLO, GLU, PHOS, and TP. The maximum and minimum values of the transformed and original data were on the same inoculation dates for ALB, ALT, CREA, CHOL, GLU, PHOS, and the A/G ratio.

### 3.6. Regression of Serum Analytes on Age at Termination

The concentrations of ALT, Ca, CREA, GLO, GLU, TBL, and TP showed significant decreasing trends with increasing age at sampling, although the odds of decrease in analyte concentrations for each month increment in age were small (odds ratios = 0.982 to 0.999). In contrast, AMYL and the A/G ratio showed significantly increasing trends by age, and odds ratios indicate that for each month increase in age, the odds of increase in AMYL concentration was 5.3%, and that for the A/G ratio it was 0.7%. Changes in the concentrations of ALB, ALKP, BUN, CHOL, GGT, and PHOS were not significant over time (Table 8). The probabilities and directions of changes in the original data for all analytes were similar to the estimates for the corresponding transformed data, except for AMYL, which was significant in the transformed data and was non-significant in the original data, although the directions of the changes over time were the same.

### 3.7. Correlation Coefficients among Serum Analytes

Pearson correlation coefficients among analytes for the transformed data are shown in Table 9. The correlation coefficient between TP and GLO (0.87) was the greatest estimate amongst all serum analytes studied. Both TP and GLO had moderate coefficients with TBL and CREA (0.46 to 0.54) and were positively and significantly correlated with other serum analytes (0.11 to 0.34), but they had no association with AMYL. Furthermore, GLO was not associated with Ca. Here, ALB had the greatest correlation coefficient with Ca (0.45), weak positive correlations with ALKP, ALT, and TBL (0.11 to 0.18, *p* < 0.05), negative correlations with GLO, BUN, CREA, CHOL, and PHOS (−0.10 to −0.19, *p* < 0.05), and no association with GGT, GLU, and AMYL.

Correlation coefficients among BUN, CREA, and PHOS, the markers of kidney function, were moderate (0.42 to 0.51, *p* < 0.01), and their correlation coefficients with TP, GLO, ALKP, GGT and, CHOL were also positive (0.11 to 0.53, *p* < 0.05), whereas their correlation coefficients with ALB were negative (−0.15 to −0.19, *p* < 0.01). Here, BUN and CREA were not associated with ALT and GLU, and CREA was not associated with Ca and AMYL. Furthermore, Ca, the other marker of kidney function, was negatively and significantly correlated with BUN, PHOS, and GLU (−0.16 to −0.28, *p* < 0.01), and positively correlated with TP and ALB (0.21 and 0.45, *p* < 0.01), but it was not associated with CREA, GLO, ALKP, ALT, GGT, CHOL, AMYL, and TBL.

The markers of liver function (ALKP, ALT, GGT, and TBL) were positively and significantly correlated with one another (0.13 to 0.35), except for GGT, which was not correlated with TBL. Ca was not associated with any of these four analytes. Here, ALKP was positively and significantly correlated with all other analytes (0.11 to 0.35). Furthermore, ALT was not associated with BUN, CREA, CHOL, and AMYL, but it was positively correlated with TP, GLO, ALB, GLU, and PHOS (0.11 to 0.21, *p* < 0.05). Here, GGT was not associated with ALB, GLU, TBL, and Ca, but was positively correlated with other analytes (0.11 to 0.30, *p* < 0.05). Additionally, TBL had moderate correlation coefficients with TP, GLO, and CREA (0.44 to 0.55, *p* < 0.01), was not correlated with BUN, GGT, and Ca, was negatively and significant correlated with AMYL, and was positively correlated with other analytes (0.11 to 0.23, *p* < 0.05).

Here, CHOL was positively and significantly correlated with most serum analytes (0.11 to 0.34), except with ALT, Ca, and GLU, with which it was not correlated, and it had a negative and weak association with ALB (*p* < 0.05). Furthermore, GLU was not correlated with ALB, GGT, BUN, CREA, CHOL, and AMYL, had a weak negative correlation with Ca and PHOS (*p* < 0.05), and had weak positive associations with TP, GLO, ALKP, ALT, and TBL (0.11 to 0.23, *p* < 0.01). Here, AMYL was not correlated with TP, GLO, ALB, ALT, CREA, GLU, and Ca, but was positively and significantly correlated with ALKP, GGT, BUN, CHOL, and PHOS (0.18 to 0.42), and was negatively correlated with TBL (*p* < 0.01). The correlation coefficients between the A/G ratio and most analytes were negative and significant (−10 to −0.88), were positive with ALB and Ca (*p* < 0.01), and A/G ratio was not correlated with ALT, GGT, GLU, and AMYL.

Of the 91 Pearson correlation coefficients, those of the original and transformed data were the same in 6 cases, the estimates were within ±0.10 point in 61 cases, and the estimates of the original data were greater than those of the transformed data between 0.11 to 0.40 points in 23 cases. In one case, the estimate of the correlation coefficient in the transformed data was greater than the corresponding value in the original data was, namely between ALB and ALKP (0.18 vs. 0.01). The greatest differences between coefficients in the two analyses involved GGT, for which the estimates of the coefficients in the original data were greater than those in the transformed data by 0.21 to 0.40 units, including the coefficients with GLO (0.52 vs. 0.12), TP (0.49 vs. 0.16), ALKP (0.63 vs. 0.35), CREA (0.37 vs. 0.14), CHOL (0.32 vs. 0.11), and TBL (0.29 vs. 0.04). Coefficients in the original data involving AMYL were also greater than the corresponding estimates in the transformed data by 0.17 to 0.28 points, including TP (0.31 vs. 0.08), GLO (0.29 vs. 0.01), CREA (0.27 vs. 0.08), PHOS (0.35 vs. 0.18), and TBL (0.12 vs. −0.14). Five of the fourteen correlation coefficients between the A/G ratio and analytes in original data (−0.16 to −0.67) had greater magnitude than those in the transformed data (−0.01 to −0.55), and the differences ranged between 0.12 and 0.29 points.

### 3.8. Correlation Coefficients between Serum Analytes and Antibody Titer

Antibody titer at pelting time showed positive and significant correlation coefficients with all serum analytes in both transformed and original data, negative and significant correlation with ALB and the A/G ratio, and was not correlated with GLU and Ca (Table 9). The greatest correlation coefficients of antibody titer were with GLO and TP in both analyses (0.49 to 0.62).

## 4. Discussion

The published estimates of selected blood analytes in mink are summarized in Appendix A. It is well established that ADV infection significantly elevates the concentrations of TP and gamma globulin (ꝩ-GLO), the major component of GLO [68], in mink [16,26,33,68,69], and the increase is more pronounced in mink that develop signs of AD [70]. These reports suggest that serum GLO and TP are valuable diagnostic tools for the evaluation of the degree of inflammation caused by AMDV infection and may aid in the identification of tolerant mink. The overall means of TP and ALB concentrations in the current study were close to those reported previously for AMDV infected mink, except in one report [68] showing very high TP (81.0 g/L) and low ALB concentrations (17.0 g/L). The TP and ALB concentrations in the current study were in the lower ranges of values reported for TP (53.0–85.3 g/L) and ALB (27.0–42.4 g/L) in non-infected mink. The large variations among reported values for TP and ALB, and the overlap between the averages of these parameters in infected and non-infected mink might have resulted from several factors, such as the time elapsed between inoculation and sampling [14,15], feed composition [52], and time of feeding [34,39], suggesting that adjustments must be made before using TP or ALB concentrations as predictors of the effects of AMDV infection on mink health. The estimates of GLO concentration and the A/G ratio in the current study were comparable to those in previous reports, namely 34.4 g/L and 0.86 [71] and 34.9 g/L and 0.81 [14].

The entrapment of immune complexes in the glomeruli of AMDV infected mink causes glomerulonephritis, interstitial nephritis, and renal dysfunction [1]. Kidneys also shown the greatest severity of AD lesions among organs in AMDV infected mink [11,14,25]. Thus, considerable changes would logically be expected in markers of kidney function, specifically CREA, BUN, PHOS, and Ca, in AMDV infected mink with renal disfunction. Indeed, CREA is a major biomarker of kidney function in some species, and is an important element in the classification of the degree of the severity of chronic kidney disease in dogs and cats as suggested by the International Renal Interest Society (IRIS) (http://www.iris-kidney.com/pdf/IRIS_Staging_of_CKD_modified_2019.pdf). The overall mean CREA concentration in the current study (82.2 μmol/L) was higher than that reported in AMDV infected mink in a previous study (46.2 μmol/L) [14], as well as in non-infected mink (38.0–82.6 μmol/L). The exceptions were for the plasma in a control group of male mink before the start of a feeding experiment (85.0 μmol/L) [39], in control female mink sampled on weeks 4, 5, and 6 of lactation and at termination (65.9–99.0 μmol/L) [72], and in lactating dams on days 35, 42, 49, and 56 of lactation, and in barren females (108.0–148.7 μmol/L) [73]. The considerable differences among CREA concentrations in different studies may be an effect of factors other than kidney damage. A large array of physiological and external factors, which are independent of the degree of renal damage, significantly influenced CREA concentrations in different species, such as time after feeding in mink [40,46], stage of lactation in mink [72,74], cooked or raw meat in mink [40] and dogs [75], dietary protein content in mink [60] and blue foxes [52], body weight in dogs and cats [48], lean body mass in dogs [76], and bacterial content of feed in mink [50]. The wide range of estimates and the effects of such a large number of factors, thus, make the merit of CREA as a predictor of kidney damage uncertain. Previous studied have also shown that CREA concentration is not affected by AMDV infection in mink [33], or by the severity of chronic kidney disease in dogs [77].

The overall mean BUN concentration in the current study (11.9 mmol/L) was higher than that reported for AMDV infected mink in a previous study (9.7 mmol/L) [14], as well as in non-infected mink (5.7 to 11.0 mmol/L). Plasma or blood urea concentrations, rather than BUN, have often been reported, and ranged between 2.4 and 17.0 mmol/L in 16 studies. The higher overall mean BUN concentration in the current study than in most published reports might have been an effect of AMDV infection, which has been found to elevate BUN concentration by more than 200% [33]. Similarly, BUN concentration in seven mink that were tested on days 43, 70, 99, and 126 dpi and showed no sign of AD have been reported to range between 4.28 and 10.7 mmol/L (12 and 30 mg/dL), and the range of the values in five mink showing signs of AD were between 8.21 and 51.4 mmol/L (23 and 144 mg/dL) [70]. The means of the values of the two groups in that study were calculated as 6.89 and 21.35 mmol/L (19.27 and 59.5 mg/dL), respectively, showing 210% higher value for those which showed signs of AD. The concentration of BUN is also positively and significantly associated with the degree of severity of chronic kidney disease in dogs based on IRIS classification [78].

The large differences amongst published reports for BUN and plasma urea concentrations might be the effects of factors, such as diet and time after feeding, in mink [40,45,46,60], blue foxes [52] and dogs [75], lean body mass in dogs [76], and lactation status in mink [74]. In an earlier experiment, ad libitum feeding following feed restriction increased BUN concentration in mink [34], whereas the plasma urea concentration was not significantly affected by feed deprivation for 7 days or re-feeding in mink [39]. It may be concluded that although AMDV infection elevates BUN concentration, the estimates require adjustment for other factors before being used as a measure of kidney dysfunction.

The overall mean PHOS concentration in the current study (2.08 mmol/L) was close to the reported values for non-infected mink (1.30 to 2.78 mmol/L), except in one report of a high value (5.9 mmol/L) for lactating mink [79]. Although impaired kidney function increased serum PHOS concentration in dogs [77,78], AMDV infection only non-significantly elevated serum PHOS concentration in mink [33], thus, suggesting that the serum PHOS concentration is not an accurate measure of tolerance in AMDV infected mink. Ca concentration in the current study (2.24 mmol/L) was lower than the values in three reports for non-infected mink (2.36 to 2.70 mmol/L). This is in agreement with a previous report indicating that AMDV infected mink have a significantly lower serum Ca concentration than healthy mink [33]. No change, however, was observed in blood Ca concentration in dogs with severe chronic kidney disease [77].

Although the liver is capable of regenerating its damaged cells [80], the severity of liver damage by AMDV infection is comparable to [11] or sometimes more extreme than that [14,81] which occurs in the kidneys. Genes that modulate liver regeneration were found to have been under selection for tolerance to AMDV infection in the same population which is used in the current study [82]. Thus, the liver appears to be an important target of AMDV infection, and several enzymes have been suggested as biomarkers of liver diseases in clinical practices, including ALT, ALKP, GGT, and TBL (reviewed in [83]).

Plasma and serum ALT activities have been proposed as biomarkers of liver damage in mink [60]. Although ALT activity is highest in the liver, it is also produced in other organs in mink [84] and blue foxes [85], thus, limiting its specificity in the blood as a biomarker of liver damage. High diagnostic sensitivity of ALT for liver diseases has been reported in humans [83] and dogs [86]. In another study in AMDV infected mink, no association was detected between the degree of liver damage and ALT activity [87], but ALT activity is elevated in the plasma of mink with high incidence of hepatic fatty infiltration [45,60]. The overall mean ALT in the current study (121.9 U/L) was lower than the 145.8 U/L in AMDV infected mink reported previously [80]. The reported means of ALT activity in non-infected mink were highly variable, showing a five-fold difference (68.8 to 385 U/L) across 12 reports. The wide range of the reported values for ALT was due to several factors affecting its activity, such as the rate of decomposition of amino acids and protein metabolism [88], the dietary protein content in mink [45,60] and blue foxes [52], the bacterial count in feed in mink [50], food deprivation in mink [39], and other factors reported for humans (reviewed in [59]). The wide range of ALT activity (21.0 to 570 U/L) and its relatively large CV (69.0) in the current study, along with large ranges of reported values for healthy mink, suggest that the prognostic value of ALT activity for tolerance to AMDV infection is uncertain.

When studied, ALKP had high activity in kidneys, followed by the intestine, and its activity in seven other organs in mink, including the liver, was low [84]. Similarly, among the eight organs in blue foxes, the highest level of ALKP has been observed in the kidneys, followed by the intestine, whereas minor activity has been observed in the liver [85]. These reports suggest that ALKP is not liver specific for mink and blue foxes. In agreement with this statement, no association was observed between the degree of liver damage in AMDV infected mink and ALKP activity [87]. In contrast, elevated ALKP activity was proposed to be indicative of liver cholestatic problems in humans [83] and biliary stasis and liver necrosis in dogs [86]. The overall mean ALKP activity in the current study (76.0 U/L) was lower than the previously reported values of 201.7 U/L [87] and 91.8 U/L in AMDV inoculated mink [14]. The activity of ALKP in non-infected mink ranged between 55.8 and 165 U/L in four reports. In another study, the serum ALKP activity in AMDV infected mink was non-significantly higher than that in healthy mink [33]. Thus, it seems that the lower overall mean ALKP activity in the current study than those in most previous reports was not due to AMDV infection.

Hepatocytes are responsible for the metabolism and excretion of bilirubin, the waste product of heme catabolism. Liver diseases, including those caused by viral infections, result in hepatocyte disfunction and hyperbilirubinemia (reviewed in [89,90]). Although AMDV infection has been reported to have no effect on TBL concentration in mink [33], the overall mean TBL in the current study (4.63 μmol/L) was greater than the 2.50 μmol/L in non-infected farmed mink and 2.035 μmol/L reported in free-ranging mink kept under the same conditions for one month prior to sampling [91]. Different levels of bacterial contamination of feed had no effect on TBL activity (2.07, 2.21, and 2.36 μmol/L) [50]. In contrast, the plasma concentrations of TBL were 7.6 and 7.7 μmol/L in male and female healthy black mink, respectively, and were not significantly changed with food deprivation [39]. Although the concentration of TBL in the plasma or serum is specific to liver problems, it is not a sensitive test for liver function, and a combination of ALT and TBL has, therefore, been suggested to be more specific and sensitive measurements than each of these analytes alone [83].

Previous studies have shown that GGT had the highest activity in the kidneys, but was not detected in the liver in mink [84] and blue foxes [85], in contrast to the suggestion that GGT is a biomarker for liver diseases in humans [83], and is indicative of biliary stasis in dogs [86]. Differences may exist among species for GGT activity in each organ, because its activity in the liver in dogs is lower than that in the organs of some other species [92]. The GGT activity in the current study (2.15 U/L) was slightly lower than that previously reported 3.02 U/L in AMDV inoculated female mink [14]. The GGT activities in two groups of mink fed bacterial contaminated diets (20.4, 22.8 U/L, corresponding to 0.34 and 0.38 μkal/L, respectively), were significantly greater than that in mink fed a diet with low bacterial contamination (4.2 U/L, 0.07 μkal/L) [50], suggesting a considerable effect of feed contamination on organ damage and GGT activity. Factors other than liver diseases influence GGT activity in different species; for example, significantly lower GGT activity has been associated with low protein diets in blue foxes [52]. Here, GGT had the greatest CV (420.8) among the analytes in the current study, partly because its activity was below the detection threshold of the test in 77.9% of mink. Similarly, GGT was not detected in the serum of some healthy dogs [92] and in dogs with liver diseases [86].

The activity of AMYL, which is produced by the pancreas and salivary glands, increases in the blood because of either an increased rate of entry into the circulation caused by pancreatic damage or decreased metabolic clearance caused by kidney failure (reviewed in [93,94]). Viral infection is among the causes of pancreatitis and the elevation of AMYL activity in the blood [95]; this was reviewed in [96]. In contrast, damage to the exocrine portion of the pancreas caused by Newcastle disease viral infection decreased the activity of AMYL and its mRNA expression in the chicken pancreas [44]. Serum and saliva AMYL activities were higher in diseased dogs than in healthy dogs [97], and both high and low AMYL activities were observed in humans with chronic kidney disease [98]. In a previous study, AMDV infection significantly increased serum AMYL activity in mink (84.0 vs. 158.0 U/L) [33], possibly as a result of either pancreatic damage or kidney dysfunction by this virus. The AMYL activity in the current study (59.0 U/L) was lower than that in healthy mink in the above study, possibly because only healthy mink were retained in the current study. In the current study, AMYL activity was below the detection limit in 82 mink, which resulted in large variation among the mink (0.0 to 339.0 U/L, CV = 92.8), suggesting the low merit of AMYL as a predictor of the state of health and degree of tolerance of the mink.

The mean GLU concentration in the current study (4.42 mmol/L) was lower than those in all previous reports (5.3–12.7 mmol/L), except in one study reporting concentrations of 3.8, 3.5, and 2.3 mmol/L during 4, 5, and 6 weeks of lactation in blush color healthy mink, respectively [72]. The AMDV infection of mink had no effect on blood GLU concentration in a previous study [33], and the low GLU concentration in the current study was caused by unknown factors. Food restriction, for instance, decreased the plasma GLU concentration in mink [34,39,99], but was not affected by the high or low protein diets in mink [45]. The overall mean CHOL in the current study (6.49 mmol/L) was within the range of values (3.98 to 9.27 mmol/L) reported in healthy mink supplemented with seven levels of copper [100], and is consistent with finding of a previous study indicating that AMDV infection had no effect on blood CHOL [33]. Plasma CHOL concentration significantly increased after food availability following food restriction in mink [34,39].

In summary, the reported estimates of blood analytes in the literature are highly variable and are each influenced by factors other than kidney or liver diseases. Differences between the overall means of serum analytes in the current study and earlier reports were primarily because the mink in the current study were inoculated with AMDV and then selected for health and productivity, and some mink had a history of selection for tolerance, thus, making this population genetically distinct from previously studied groups. In addition, the animals were sedated prior to sampling, samples were processed the day after sampling, stored at −80 °C, and tested at later dates with different laboratory methods. These factors might explain the observed differences between the estimates in the current study and previously reported values. However, the overall means of all blood analytes in the current study were within the range of reported values, although some analytes generally had higher (CREA, BUN) or lower (Ca, GGT, AMYL, GLU) estimates than most reported values.

It was observed that concentrations of GLO and TP in the transformed data linearly decreased as the percentage of tolerance ancestry increased, thus, resulting in the significantly lower concentrations of these analytes in the tolerant mink compared to the unselected mink (Table 3). This was a result of the previous selection for tolerance. It may be hypothesized that the tolerant mink were genetically capable of suppressing production of GLO after infection. Despite significant differences in the concentrations of GLO and TP among tolerant groups, the concentrations of ALB were comparable among tolerant groups, a result of the balancing effect of ALB in maintaining the colloidal osmotic pressure of the blood [101]. The linear and significant increases in the A/G ratio, which was parallel to the increasing tolerance ancestry, resulted from decreasing GLO and no change in ALB concentration by increasing tolerance ancestry. There were lower values of BUN, CREA, PHOS, ALT, ALKP, and GGT in TG100 or TG75 compared to TG0 or TG50, although differences were mostly not significant, which might suggest a tendency toward improved renal function as a result of previous selection for tolerance. Because GGT and ALKP have the highest activities in the kidneys among all organs in mink [84], they may not be robust biomarkers of liver function in mink, which is contradictory to the previously suggested biomarkers for the liver in humans and dogs [83,86]. The significantly lower Ca concentration in TG50 than in TG75, and the opposite ranking of the tolerant groups for PHOS concentrations, might also be indicative of the improved renal function as a result of selection for tolerance.

The results of the original and transformed data for the tolerant groups were comparable for most analytes, implying that transformation was not required for most analytes. Significant levels changed for GGT and PHOS, which were significant only in the original data. The rank order of the least-squares means and back-transformed values were the same for all analytes, except ALKP, ALT, AMYL, BUN, CREA, and GGT, in which the position of only one tolerant group differed between the two analyses. The maximum and minimum values were found in the same tolerant groups for all analytes, except for ALKP, AMYL, and BUN, for which only the maximum or minimum values were in different tolerance groups. These differences were, thus, infrequent and had minor effects on the interpretation of the results.

The finding that the greatest correlation coefficients between antibody titer and serum analytes were for TP (0.53), GLO (0.62), and A/G (−0.60) (Table 9) suggested their strong associations with the degree of tolerance of mink to AMDV infection. These coefficients were comparable to those between antibody titer and TP (0.51), GLO (0.57), and A/G (−0.53) in AMDV-inoculated mink sampled at 451 dpi in a previous study [14]. As expected, the correlation coefficients between serum ꝩ-GLO and antibody titer in previous studies were greater than those between antibody titer and GLO in the current study, namely 0.81 [16], 0.61 [102], and 0.75 [29]. The weak but positive and significant correlation coefficients between antibody titer and most other analytes (ALKP, ALT, GGT, BUN, CREA, CHOL, PHOS, AMYL, and TBL) indicate that they were not robust diagnostic tools for the estimation of the degree of tolerance of mink. In a previous study, the Spearman rank correlation between antibody titer and serum ALKP, BUN, CREA, and GGT concentrations in AMDV-inoculated mink were negligible (−0.03 to −0.09) [14], supporting the notion that these analytes have low diagnostic power for estimating the degree of tolerance. The negative association between ALB concentration and antibody titer in the current study (−0.17) and a previous report (−0.21) [14] is the response of mink to increased antibody titer and GLO concentration in the blood for maintaining blood colloidal osmotic pressure. The similarity between the estimates of correlation coefficients of transformed and original data suggests the minor effects of deviation from normality on correlation coefficients.

The highest correlation coefficient among serum analytes was between GLO and TP in the current study (0.87) and a previous study (0.88) [14] because GLO constitutes the major component of TP. The positive and significant correlation coefficients of TP and GLO with most other analytes, which were largest with CREA and TBL, suggested the elevation of the concentrations of the serum analytes in response to kidney and liver functions. In a previous study, TP and GLO were also positively correlated with BUN and CREA (0.16 to 0.25) in AMDV inoculated mink [14]. The negative correlation coefficients between GLO and ALB in the current (−0.10) and a previous study (−0.29) [14], is the result of decreased ALB concentration after increased ꝩ-GLO [68] or GLO concentrations [33] in AMDV infected mink. This finding was a result of diminished ALB synthesis by the liver to maintain blood colloidal osmotic pressure (reviewed in [101]). In an earlier study in AMDV inoculated mink [14], ALB concentration had a negative and mostly non-significant Spearman rank correlation with BUN, CREA, GLO, ALKP, and GGT (−0.02 to 0.29).

The positive and moderate correlations among BUN, CREA, and PHOS (0.42 to 0.51) suggest their analogous responses to the degree of kidney function. The positive and significant correlations of BUN, CREA, and PHOS with GLO, TP, ALKP, and GGT (0.11 to 0.53) and their negative and significant correlation with ALB concentration (−0.15 to −0.19) confirmed the response of these analytes to liver function and general health as well. Previous studies in dogs have revealed that the correlation coefficient between CREA and BUN concentrations was 0.52 [76], a value comparable to the estimate in the current study (0.47), and that between CREA and serum urea the correlation coefficient was 0.79 [103]. In another study [14], the Spearman rank correlation coefficient between BUN and CREA in AMDV-inoculated mink was positive and large (0.85), whereas those between BUN and CREA with ALB, ALKP, GLO, TP, and GGT were small and non-significant (−0.01 to 0.25).

The negative and significant correlation between PHOS and Ca (−0.27), and the observation that PHOS concentration was positively and significantly correlated with BUN (0.51) and CREA (0.42), the other two markers of kidney function, whereas the corresponding coefficients between Ca with BUN (−0.28) and CREA (−0.08) were negative, suggested an antagonistic effect between animal health and PHOS and Ca concentrations. This finding is in line with an earlier report indicating that AMDV infection non-significantly elevated serum PHOS concentration but significantly decreased serum Ca concentration in mink [33]. The converse effects of Ca and PHOS concentrations on other serum analytes, namely the positive and significant correlations between PHOS concentration and other analytes, except ALB and GLU, and the negative associations of Ca with those analytes, and the finding that ALB had the greatest positive correlation with Ca (0.47) but a negative and significant correlation with PHOS (−0.19), are the manifestation of Ca-PHOS homeostasis.

The positive and significant correlations among ALKP, ALT, GGT, and TBL, except GGT and TBL which were not correlated, implied that these analytes have a parallel response to the degree of liver function. Here, ALKP, GGT, and TBL had positive and small to moderate (0.12 to 0.55, *p* < 0.05) correlation coefficients with TP, GLO, BUN, CREA, PHOS, and AMYL, suggesting that they have comparable effects on kidney function and animal health as well. In contrast, AMDV infection significantly increased BUN, TP, GLO, and AMYL, but the increase in ALKP activity was not significant [33]. In a previous study [14] a positive and significant Spearman rank correlation between GGT and ALKP (0.33) was observed in AMDV inoculated mink, whereas ALKP and GGT had weak associations with TP, ALB, BUN, CREA, and A/G ratio (−0.14 to 0.20). Positive association between GGT and ALKP was also observed in dogs [92], and a significant correlation coefficient between ALKP and GGT was reported in dogs with liver diseases (0.58) [86]. The ALT concentration, in contrast, was not a strong predictor of kidney functions because it was not associated with BUN or CREA.

The positive and significant correlation coefficients between CHOL and most analytes, which were greatest with TP (0.29) and GLO (0.31), are indicative of the modest effects of the liver and kidney functions and general health on CHOL concentration. In a previous study, AMDV inoculated mink have been found to show significant increases in TP, BUN, GLO, and AMYL concentrations, but a non-significant increase in CHOL concentration [33], in agreements with the small to moderate positive correlations between CHOL and most other analytes observed in the current study. The finding that GLU was not correlated with ALB, GGT, BUN, CREA, CHOL, and AMY, and had small correlation coefficients with other analytes (−0.16 to 0.23) suggests its low merit as a diagnostic tool for health and tolerance in mink. The finding is in agreement with results from a previous report indicating that AMDV infection has no effect on blood GLU concentration in mink, whereas BUN, TP, GLO, and AMYL are significantly elevated [33].

The positive and significant, but mostly small, correlation coefficients between AMYL and ALKP, GGT, BUN, and PHOS, and the absence of association with TP, GLO, or ALB concentrations, may suggest that changes in AMYL concentration were primarily the result of pancreatic damage rather than kidney dysfunction. This conclusion was based on previous reports indicating that AMYL activity is influenced by damages to the pancreas and kidneys [93,94,98]. However it was not associated with TP, GLO, ALB, and CREA in the current study. In contrast, serum AMYL level was significantly correlated with serum urea (0.23), CREA (0.17), TP (0.27), and ALB (0.41) in humans [98]. The AMYL, BUN, TP, and GLO concentrations are significantly higher in AMDV infected than healthy mink [33], and serum AMYL activity is higher in diseased than in healthy dogs [97]. It is interesting to note that AMYL was the only analyte not associated with TP concentration in the present work. The observation that differences between correlation coefficients in the transformed and original data were small in 67 of the 91 comparisons indicated that the effect of transformation on correlation coefficients was modest, and the greatest differences involved GGT and AMYL which had zero values and high deviations from normality.

The significantly higher concentration of PHOS for males than females in the original and transformed data is in agreement with a finding in a previous report in mink [47], possibly as a result of an association of PHOS with sex hormones [104,105]. In contrast, no sex difference has been reported for PHOS in mink [106], dogs [42,107], and prairie dogs [108]. The significantly higher concentration of TBL in males than females in both the transformed and original data might have been associated with the relationship between TBL and hemoglobin metabolism [109], which is higher in males than females because the number of red blood cells (packed cell volume, erythrocytes, and hematocrit) and the hemoglobin concentration are significantly greater in male than in female mink [39,47,49], rats [41], and beech martens [53]. The difference between sexes have also been attributed to the higher oxygen binding capacity for male rather than female beech martens [53]. Contrary to the results of the current study, the concentration of TBL in male dogs is significantly lower than that in females [42], but the concentrations of TBL in male and female mink [39] and prairie dogs [108] do not differ.

Male mink tended to have a higher CREA concentrations than females in the original and transformed data, which agrees with the significantly higher CREA concentrations in male than female dogs [42,107]. The higher concentration of CREA in male than female dogs has been attributed to the higher body weight and muscle mass of males [48,76,103]. The significantly greater body mass of male than female mink [39] might have been a factor contributing to the observed difference between sexes in the current study. Contrary to these findings, CREA concentrations in prairie dogs [108] and rats [41] were significantly higher in females than males, but no sex effect was reported for mink [39,106], healthy dogs and cats [48], and beech martens [53].

The finding that CHOL was the only analyte with a significantly lower concentration in males than females in both the original and transformed data was likely associated with the effects of sex hormones, because CHOL is the precursor molecule for the sex hormones (progesterone, estrogen, estradiol, and testosterone) [110], and sex hormone concentrations have positive and significant associations with CHOL concentration in humans [111,112]. The significantly greater concentration of CHOL in females than males is consistent with reports in rats [41] and dogs [113]. In another study, the plasma concentrations of CHOL were not significantly different between sexes in fasted and re-fed mink [39].

Males had greater ALT activity than females in the transformed (*p* > 0.05) and original (*p* < 0.05) data in the present study, which agrees with the significantly higher ALT activity in male compared to female mink [47], dogs [42], prairie dogs [108], rats [41], and humans [59,114]. In previous studies, however, differences between male and female mink [106] and beech martens [53] were not statistically significant. In another study, a significant fasting by sex interaction was detected, where male mink had lower plasma ALT activity than females before and after 5 and 7 days of fasting, but the ALT activity in males was greater than that in females after one and three days of fasting and at 28 days after re-feeding [39].

In agreement with the results of the current experiment in the transformed and original data, most previous reports have shown no significant sex effects for TP and ALB concentrations. The two sexes have comparable concentrations of TP in mink [47,106], rats [41], prairie dogs [108], and beech martens [53]. In other studies, the concentrations of plasma TP were significantly higher in males than females in mink [39] and dogs [42,107]. The two sexes have comparable concentrations of ALB in mink [106], rats [41], dogs [107,113], and beech martens [53], whereas females have significantly higher plasma concentration of ALB than males in dogs [42] and prairie dogs [108]. In contrast, the concentrations of serum ALB in male mink were significantly greater than those in females [47]. In agreement with the result of the current study in the transformed data, male dogs had a significantly higher plasma concentration of GLO than females [42,107], but the opposite was reported for rats [41]. The similarity between sexes for GLO concentration in the original data was consistent with that in previous reports in prairie dogs [108] and ꝩ-GLO in beech marten [53].

The finding that Ca concentration was not affected by sex in the original and transformed data agrees with earlier reports in mink [106], dogs [42,107], and prairie dogs [108], whereas the serum Ca concentration was significantly lower in female than male mink [47]. Furthermore, BUN and AMYL were the only analytes whose concentrations were greater in females than males, but the differences were significant only in the transformed data. In previous studies, serum urea concentrations were significantly greater in female than male rats [41] and dogs [113]. In contrast, significantly greater concentrations of AMYL in male than female dogs [107], and concentrations of BUN in dogs [42], were reported. In agreement with the results of the original data, no difference between males and females was reported for the plasma concentration of AMYL and BUN in prairie dogs [108]. Moreover, no difference between sexes in the concentrations of urea in the plasma and serum in mink [39,47], or for the concentration of BUN in dogs [76,107], healthy dogs and cats [48], and beech martens [53], have been reported. Similarities between males and females in ALKP, GGT, and GLU estimates in both the transformed and original data were in agreement with reports of GLU concentrations in male and female mink [39,106], dogs [107,113], and prairie dogs [108], and with ALKP activity in dogs [107] and prairie dogs [108]. In contrast, significantly higher concentrations of GLU have been observed in females than males in dogs [42] and rats [41], whereas significantly lower concentrations have been found in females than males in mink [47]. Moreover, significantly higher ALKP activity has been observed in male compared to female dogs [42].

The substantial inconsistencies in the difference between sexes in the published reports in different species for each analyte, and the differences between the results of the current study and published findings in mink, are suggestive of the contributions of many factors, such as sex hormones, health status, age, reproductive stage, muscle mass, and body fat content, among others, on analyte concentrations. Thus, it seems logical to suggest that the concentrations of serum analytes must be measured within each sex to make informed judgements regarding the merit of each as an indicator of health status and tolerance in mink.

Although Box–Cox transformation substantially changed the shapes of the distributions of the serum analytes, differences between sexes in the transformed and original data were comparable for most analytes. The probabilities of 11 of the 14 analytes, and the ranking of the back-transformed values and least-squares means were the same for all analytes except 1 (Ca). The presence of only two sex groups, which resulted in a large number of observations within each group, might have been the reason for the small differences between the results of the analyses of variance.

Inoculation date was included in the statistical model to account for the effects of unknown physiological and environmental factors on the concentrations of the analytes. Several factors might have caused most analyte concentrations to significantly differ between inoculation dates. First, the physiological status of the mink might have varied among years and seasons, thus, affecting some serum analyte concentrations. In an earlier study, sampling year had significant effects on serum concentrations of GLU, TP, ALB, GLO, Ca, PHOS, CREA, TBL, BUN, and the activities of ALKP and ALT in endurance-trained sled dogs sampled over 7 years [42]. In other studies, the plasma activity of ALT increased in mink tested five times between July and December, whereas concentrations of BUN and GLU did not show clear patterns of changes over time [45]. The concentrations of TP and urea in mink plasma significantly decreased from September/October to December, whereas the plasma activity of ALT moved in an opposite direction [60], and plasma ALB and TP concentrations increased in mink measured in July, September, October, and December [51].

Second, the mink inoculated at later dates in the current study were the progeny of those that had been selected for health and productivity and were, thus, genetically different from those inoculated at earlier times. This process is commonly used on commercial mink farms. Third, blood processing procedures might have significantly affected the concentrations of some analytes. Although the length of time between sample collection and processing affects the concentrations of some analytes (reviewed in [115]), keeping this period short and uniform for all mink was not practical in the current study because a large number of mink were euthanized each day. Blood samples were kept in a refrigerator at the barn for up to five hours, transported 50 km to the laboratory in a cooler, and stored in a refrigerator overnight before separation of the serum. Significant differences among inoculation dates for some blood analytes might have partly been the result of differential responses of analytes to the blood handling conditions [116]. In addition, the mink in the current study were terminated in January or February when the outside temperature was low, with day-to-day and year-to-year fluctuations. Although the temperature inside the mink barn was above freezing, it was influenced by the outside temperature, thus, potentially affecting the concentrations of some analytes. Although the sample processing conditions were not uniform for all mink, similar situations would not be provided at any commercial mink farm. Fourth, animals were terminated between 120 and 1211 dpi during 2011 and 2014, and serum samples were frozen and tested in 2013 to 2015; thus, a combination of the age of the mink and duration of storage could potentially have affected the concentrations of some analytes. Older animals at termination were those which were inoculated in 2010 or 2011, and their serum samples were stored for as many as 800 days at −80 °C before testing. The long-term storage of serum and plasma sample at −80 °C has been recommended for the maximum stability of analytes [117], although concentrations of some analytes changed at this temperature [118,119].

Amongst the 14 analytes tested in the current study, inoculation date had no effect on the concentrations of BUN, Ca, PHOS, and GGT in the transformed and original data. The absence of the significant effect of inoculation dates on Ca and PHOS might be associated with the stability of Ca [54,118,120,121] and PHOS [54] after storage at −80 °C. An earlier study has also shown that the activity of GGT is not affected by 1 year of storage at −80 °C [119,122], although its activity increased after 7 and 10 years of storage [119], probably because of water evaporation. The effects of the duration of storage on the concentrations of BUN in earlier studies have been inconsistent, showing no change after 1 year [54] and 10 years [119] of storage, significantly increased after 7 days [118] and 7 years [119], or decreasing after 1 year of storage [119].

The significant decreasing trends in the concentrations of CREA, CHOL, and GLO with advancing inoculation dates in both the transformed and original data were probably not an effect of the duration of storage, because most previous studies have indicated the stability of these analytes during storage. Previous studies have shown no changes in the concentrations of CREA after 7 days [118], 1 year or 10 years [119], and no change in the concentration of CHOL after storage for 7 days [118], 90 days [121], or 10 years [119]. Significant increases in the concentration of CREA after 7 years of storage [119] and an increase in the concentration of CHOL after 1 and 7 years of storage [119] have also been reported, thus, contradicting the notion that storage does not have effects on these analytes, as well as the significant decrease in the concentrations of CREA and CHOL observed in the current study. There was a significantly lower concentration of TBL in samples from mink that were inoculated in 2013 rather than on earlier occasions, and were, thus, stored for a longer time, which indicated that factors other than the duration of storage might have caused the increase in its concentration, a finding contrary to those in earlier studies showing that storage of serum or plasma for 7 days, 1, 2, 3, 4, or 6 months, and 1, 7, or 10 years at −70 °C or −80 °C significantly decreases the concentration of TBL [118,119,123].

The increasing trends in the concentrations of ALB, ALT, GLU, and the A/G ratio, which resulted in significantly smaller values for mink inoculated in October 2010 than in September 2013, might have partly been the effects of the duration of storage, which has been found to have inconsistent effects on these analytes in previous studies. The activity of ALT did not change [54,118,121] or decreased [114,119,122] with storage, the concentration of ALB did not change [54,118,121,123], increased [119,123], or decreased after stored for 7 or 10 years [119], and the concentration of GLU did not change after storage between 90 days and 10 years [54,119,121]. The finding that concentrations of ALKP, AMYL, and TP were significantly affected by inoculation dates but did not follow any specific trend suggested minimal effects of the duration of storage on these analytes. In previous studies, ALKP, AMYL, and TPP have been found to be stable after storage between 7 days and 1 year [54,118,121,123] but, in another study, the concentration of TP increased after 2 to 6 months of storage [123]. It may be concluded that a wide array of environmental, technological, and physiological factors influences the concentrations of analytes, and need to be controlled in order to be useful predictors of mink health and tolerance to infection. Some degree of control, such as over the season and month of sampling and duration of storage before sample analysis, is possible. It must be noted that BUN, Ca, and GGT appear to be the most stable analytes because their concentrations did not change with inoculation date.

The probabilities of differences among inoculation dates were the same for the transformed and original data, except for ALKP, which was significant only in the transformed data, suggesting that the transformation had minor effects on the interpretation of the results. The rank orders of the back-transformed values were the same as the least-squares means of the original data for ALB, Ca, CREA, CHOL, TBL, and the A/G ratio, but differences were observed in the rank orders of the other analytes. The maximum and/or minimum values were the same in both analyses, except for BUN and GGT, thus, supporting the statement that the effects of transformation on the interpretation of the results were trivial.

The process of aging causes great changes in the biological and physiological conditions of animals, including the profiles of blood analytes. To our knowledge, no published report has described the effects of age longer than 6 months on blood analytes in mink. Previous studies in dogs showed significant, often non-linear, changes in blood analyte with age, and sometimes with an interaction with sex [42,107] or breed [57]. In addition, most changes occurred in juveniles, whereas analyte concentrations become relatively stable in healthy older dogs [57,107], suggesting that the magnitude and direction of change in blood analytes depend on the ranges of the distribution of age in each study. The finding that age did not have a significant effect on the concentrations of ALB, ALKP, BUN, CHOL, GGT, and PHOS in the transformed data in the current study might have been partly because of the rather narrow range of age (120 to 1211 dpi), in agreement with the findings of no change with age for ALKP, CHOL, and PHOS in adult dogs [57], PHOS in prairie dogs [108], BUN in dogs [42,113], plasma urea in adult dogs [57], and serum urea in cats [43]. In contrast, concentrations of these analytes significantly decreased with age in other studies, namely ALB concentration in dogs [42,107], prairie dogs [108], and humans [59], ALKP in dogs [42,113] and prairie dogs [108], GGT in humans [59], PHOS in dogs [42,107], and BUN in prairie dogs [108], and female dogs [113], whereas concentrations of ALB and CHOL significantly increased with age in dogs [113].

The significant decreases in the concentrations of ALT, Ca, CREA, GLO, GLU, TBL, and TP with age in the transformed and original data agree with the effects of age on GLU in dogs [42,113] and prairie dogs [108], Ca in dogs [42,107], and ALT activity in prairie dogs [108] and humans [59]. In contrast, significant increases with age were reported for CREA concentration in prairie dogs [108], GLO in dogs [42,107] and prairie dogs [108], TP in dogs [42,107,113] and prairie dogs [108], ALT in dogs [107], and TBL in dogs [42] and humans [59]. The TBL concentration increased in female dogs up to 2 to 4 years of age, remained at a plateau until 6 to 8 years of age, and decreased in dogs older than 10 years of age [107]. Other studies have shown no significant changes in the concentrations of CREA in dogs [42] and cats [43], Ca concentrations in adult dogs [57,107] and prairie dogs [108], GLO in adult dogs [57], and TP and ALT in dogs [42,57]. Limited information is available on the effect of age on AMYL concentration, which significantly increased with age in the transformed data but was not affected by age in the original data. The AMYL activity does not change with age in prairie dogs [108], but is high in young female dogs, decreases up to the ages of 4 to 6 years, and then increases again in dogs older than 10 years [107]. The inconsistent, often contradictory, patterns of changes in the concentrations of analytes in current and published reports imply that adjustments for age are required when analyte values are used as biomarkers of health and tolerance in mink.

Large differences in the shapes of the distributions were observed among the 14 analytes, as indicated by differences between the means and medians of each distribution (Table 1) and for the estimates of skewness and kurtosis (Table 2). All analytes significantly deviated from normality, thus, providing an opportunity to relate the shape of the distributions to the need for data transformation and the effects of transformation on the results of analysis of variance and mean comparison. In agreement with previous reports [35,114], 12 of the 14 serum analytes in the current study were positively skewed, and their means were greater than their medians. The positive estimates of skewness were due to several exceptionally large values caused by physiological and exogenous factors, such as diseases [79], diet composition [34,45,60,124], and sampling time [45,60]. Nursing sickness, for example, resulted in 64%, 76%, 342%, and 890% greater concentrations of PHOS, CREA, GLU and urea, respectively, than those in healthy nursing dams [79]. The Box–Cox power parameter (λ) for the three negatively skewed distributions (ALB, Ca, and A/G) was greater than 1.0, whereas the estimate of λ for other analytes was smaller than 1.0 and mostly negative. Logarithmic transformation (λ = 0), which is often used for transforming blood analytes, was not optimal for the serum analytes, except for BUN and CREA. Although the Box–Cox transformation moved the distributions closer to normality, as shown by the closer estimates of skewness and kurtosis to zero (Table 2), the transformed data of the analytes, except those of CREA and GLU, remained significantly deviated from normality, suggesting that Box–Cox transformation shifts distributions closer to normality but does not ensure normality.

Significance levels of ALB, Ca, CREA, CHOL, GLO, GLU, TBL, TP, and the A/G ratio were the same for all parameters in the statistical models (tolerance, sex, inoculation date, and regression on age) in the original and transformed data. These analytes had negative skewness (ALB, Ca, and the A/G ratio) or their estimates of skewness were positive but small (0.93 to 1.77). Here, AMYL was an exception because its skewness was smaller than 1.77 but the probabilities of the original and transformed data differed for tolerance groups, inoculation date, and regression on age. Interestingly, except for AMYL, all analytes with skewness greater than 1.77 had only one parameter that significantly differed between the transformed and original data, namely the probabilities for GGT and PHOS were different for the effect of tolerance groups, ALT and BUN were different for sex, and ALKP was different for inoculation date. It may be concluded that transformation of blood analytes may not generally be necessary when their skewness is negative or smaller than 1.8, and no difference may exist between the results of analysis of variance of the original and transformed data with skewness greater than 1.8 for most parameters in the statistical models.

The other important consideration regarding the effects of transformation on the results of the analysis of variance is the similarity of the rank orders of the means and statistical differences of the least-squares means in the original and transformed data. With minor differences, the rank orders of the least-squares means and back-transformed values were the same for the analytes that had the same probability levels in the two analyses, i.e., ALB, Ca, CREA, CHOL, GLO, GLU, TBL, TP, and A/G. Differences included the opposite ranking of males and females for Ca and the different order of one of the means for CREA. Finally, the magnitudes of least-squares means of the original data were greater than the back-transformed values for most analytes in all classification parameters, and the differences were greatest for GGT and AMYL. These analytes contained several zero values, which were converted to 0.01 before transformation. Raising values smaller than 1.0 to a power reduces the results [125]. The least-squares mean of the transformed data were back-transformed without any adjustment, as recommended for reducing the transformation bias [61,65,66], but differences between the magnitudes of back-transformed values and least-squares means appeared to have had little effect on the interpretation of the results.

## 5. Conclusions

The results of the analysis of serum analytes in the current study and published reports in carnivores and rodents indicated that a large array of factors, including species, age, sex, feed composition, nutrient quality, time of feeding, sample preparation methods, and duration of sample storage, significantly affected the concentrations of serum analytes. It was concluded that the efficacy of serum analytes as biomarkers of mink health and tolerance to AMDV infection is limited. Concentrations of most analytes were lower in mink that had been selected for tolerance than in unselected mink, but the differences were mostly minor, confirming their low merit as the biomarkers of tolerance. Total serum protein and globulin concentrations were the most meaningful analytes as biomarkers of tolerance because they were significantly lower in the selected than the non-selected mink and had the greatest correlation coefficient with antibody titer. Distributions of all analytes significantly deviated from normality, and data were analyzed after Box–Cox power transformation. The probabilities of differences among the levels of the parameters in analyses of variance and the ranking orders of least-squares means of the parameters in the original data and back-transformed values were the same in the two sets of analyses for most analytes, suggesting that data transformation prior to statistical analysis was needed only for those analytes whose estimate of skewness were positive and greater than 1.8.

## Figures and Tables

**Table 1 animals-12-02725-t001:** Descriptive statistics of serum analytes at termination.

Analyte ^§^	Number	Mean	Median	Range	CV ^¥^
ALB, g/L	493	27.7	28.0	15.0–36.0	8.7
ALKP, U/L	493	76.0	67.0	25.0–369.0	49.9
ALT, U/L	413	121.9	100.0	21.0–570.0	69.0
AMYL, U/L	413	59.0	51.0	0.0–339.0	92.8
BUN, mmol/L	493	11.9	11.0	0.31–46.4	53.8
Ca, mmol/L	413	2.24	2.25	1.69–2.83	5.3
CREA, μmol/L	493	82.2	75.0	19.0–286.0	44.1
CHOL, mmol/L	413	6.49	6.34	3.50–10.70	17.9
GGT, U/L	408	2.15	0.0	0.0–72.0	420.8
GLO, g/L	493	32.4	30.0	22.0–60.0	21.7
GLU, mmol/L	413	4.42	4.35	0.63–11.20	38.6
PHOS, mmol/L	413	2.08	1.95	1.08–5.19	30.6
TBL, μmol/L	413	4.63	4.00	2.0–14.5	57.5
TP, g/L	493	60.0	58.0	45.0–89.0	11.9
A/G	493	0.89	0.92	0.44–1.44	19.9

^§^ Abbreviations are as follows: LB, albumin; ALKP, alkaline phosphatase; ALT, alanine aminotransferase; AMYL, amylase; BUN, serum urea nitrogen; Ca, calcium; CREA, creatinine; CHOL, cholesterol; GGT, gamma-glutamyl transferase; GLO, globulin; GLU, glucose; PHOS, phosphorus; TBL, total bilirubin; TP, total protein, A/G, ALB/GLO ratio. ^¥^ Coefficient of variation.

**Table 2 animals-12-02725-t002:** Skewness and kurtosis of the original and Box–Cox transformed data and the Box–Cox power parameters (λ).

	Original Data			Box–Cox Transformed Data	
Analyte	Skewness	Kurtosis	λ	Skewness	Kurtosis
ALB	−0.64	1.86	2.75	0.08	0.92
ALKP	3.58	18.98	−0.75	−0.12	0.85
ALT	2.56	8.20	−0.25	0.08	0.48
AMYL	1.49	3.57	0.25	−0.90	−0.47
BUN	2.78	10.38	0.0	0.14	1.92
Ca	−0.13	2.50	1.5	0.04	2.58
CREA	1.55	4.25	0.0	0.00	0.29
CHOL	0.97	1.56	−0.5	−0.04	0.89
GGT	5.91	37.06	−0.75	1.37	−0.12
GLO	1.77	3.01	−2.25	0.07	0.14
GLU	0.62	1.04	0.5	−0.11	0.44
PHOS	2.50	8.52	−1.0	0.00	0.87
TBL	0.93	0.48	−0.25	−0.02	−1.44
TP	1.52	2.80	−2.75	−0.07	1.11

**Table 3 animals-12-02725-t003:** Back-transformed values of blood analytes of the tolerant groups and the probabilities of differences among groups (Prob) ^§,¥^.

Analyte	TG0	TG50	TG75	TG100	Prob
ALB	27.75	27.64	27.97	28.01	0.71
ALKP	72.51 ab	74.43 a	53.75 b	65.76 ab	0.002
ALT	104.12	96.17	73.85	94.57	0.11
AMYL	29.56	19.47	47.98	21.04	0.23
BUN	11.72 ab	11.65 ab	12.41 b	10.07 a	0.01
Ca	2.22 ab	2.18 a	2.26 b	2.25 ab	0.03
CREA	76.52	79.56	82.53	76.44	0.82
CHOL	6.50	6.44	6.77	6.36	0.42
GGT	0.016	0.012	0.012	0.013	0.25
GLO	33.07 a	30.56 ab	30.42 ab	29.91 b	0.001
GLU	4.56	4.82	3.95	4.18	0.11
PHOS	2.04	2.10	1.95	1.93	0.14
TBL	4.19	4.80	3.81	4.00	0.28
TP	61.57 a	58.33 ab	58.75 ab	57.92 b	0.001
A/G	0.83 a	0.89 ab	0.91 ab	0.93 b	0.001

^§^ Means followed by different letters in rows are different at *p* < 0.05. ^¥^ In this and other tables, probabilities and mean comparisons are based on the results of the analysis of variance of the transformed data, and back-transformed least-squares means.

**Table 4 animals-12-02725-t004:** Least-squares means ± standard errors of the original serum analytes for the tolerant groups and the significance levels of differences among means (Prob) ^§^.

Analyte	TG0	TG50	TG75	TG100	Prob
ALB	27.62 ± 0.23	27.49 ± 0.48	27.81 ± 0.69	27.84 ± 0.16	0.74
ALKP	83.79 ± 3.79 a	82.48 ± 7.75 ab	59.57 ± 11.21 b	72.45 ± 2.59 b	0.03
ALT	124.83 ± 8.76	114.11 ± 16.74	66.02 ± 24.06	114.56 ± 6.27	0.13
AMYL	66.93 ± 5.86	53.26 ± 11.21	75.75 ± 16.11	52.95 ± 4.19	0.12
BUN	13.42 ± 0.63 a	13.59 ± 1.29 a	13.14 ± 1.87 ab	10.91 ± 0.43 b	0.003
Ca	2.22 ± 0.01 ab	2.18 ± 0.02 a	2.26 ± 0.03 b	2.25 ± 0.01 b	0.03
CREA	86.03 ± 2.88	85.25 ± 5.90	86.39 ± 8.53	81.22 ± 1.97	0.49
CHOL	6.68 ± 0.11	6.63 ± 0.22	6.83 ± 0.31	6.50 ± 0.08	0.42
GGT	4.74 ± 0.96 a	1.14 ± 1.88 b	2.26 ± 2.65 ab	1.18 ± 0.69 b	0.02
GLO	35.72 ± 0.55 a	32.06 ± 1.12 b	31.69 ± 1.63 b	30.70 ± 0.37 b	0.001
GLU	4.71 ± 0.18	4.95 ± 0.34	3.98 ± 0.49	4.32 ± 0.13	0.07
PHOS	2.25 ± 0.07 ab	2.33 ± 0.13 a	2.02 ± 0.18 b	2.01 ± 0.05 b	0.007
TBL	5.09 ± 0.24	5.74 ± 0.46	4.37 ± 0.68	4.73 ± 0.17	0.11
TP	63.33 ± 0.59 a	59.58 ± 1.21 ab	59.62 ± 1.75 ab	58.53 ± 0.40 b	0.001
A/G	0.81 ± 0.01 a	0.88 ± 0.03 ab	0.90 ± 0.04 ab	0.92 ± 0.01 b	0.001

^§^ Means followed by different letters in rows are different at *p* < 0.05.

**Table 5 animals-12-02725-t005:** The back-transformed values of Box–Cox transformed data and least-squares means ± standard errors of the original serum analytes of males and females, and probabilities of differences between sexes (Prob).

Analyte	Transformed Data			Original Data		
	Female	Male	Prob	Female	Male	Prob
ALB	27.72	27.96	0.27	27.61 ± 0.25	27.75 ± 0.27	0.52
ALKP	65.56	65.78	0.91	73.31 ± 4.08	75.80 ± 4.40	0.49
ALT	87.70	95.00	0.11	95.79 ± 9.17	113.99 ± 9.86	0.03
AMYL	38.62	19.83	0.001	65.17 ± 6.15	59.88 ± 6.6	0.29
BUN	11.98	10.90	0.026	12.89 ± 0.68	12.64 ± 0.73	0.67
Ca	2.24	2.22	0.32	2.23 ± 0.01	2.28 ± 0.01	0.31
CREA	76.60	80.90	0.08	82.07 ± 3.11	87.37 ± 3.34	0.06
CHOL	6.83	6.22	0.001	6.94 ± 0.12	6.37 ± 0.13	0.001
GGT	0.013	0.014	0.20	1.71 ± 1.07	2.95 ± 1.09	0.18
GLO	30.56	31.25	0.06	32.29 ± 0.59	32.80 ± 0.64	0.33
GLU	4.32	4.41	0.59	4.42 ± 0.19	4.56 ± 0.20	0.39
PHOS	1.94	2.07	0.008	2.08 ± 0.07	2.23 ± 0.08	0.02
TBL	3.85	4.54	0.001	4.71 ± 0.26	5.26 ± 0.27	0.02
TP	57.12	61.57	0.35	59.95 ± 0.63	60.58 ± 0.68	0.26
A/G	0.90	0.88	0.26	0.89 ± 0.01	0.87 ± 0.02	0.29

**Table 6 animals-12-02725-t006:** Back-transformed values of serum analytes for the inoculation dates and the probabilities of differences among groups ^§^.

Analyte	7–18 Oct.	13 Dec.	13–20 Sept.	13 Dec.	11–20 Sept.	10–17 Sept.	Prob
	2010	2010	2011	2011	2012	2013	
ALB	26.89 a	27.46 ab	28.26 b	27.84 ab	28.12 b	28.43 b	0.001
ALKP	59.94 a	62.60 ab	61.43 a	72.86 b	67.25 ab	71.97 b	0.001
ALT	77.69 a	83.74 ab	92.16 ab	88.32 ab	97.49 ab	113.05 b	0.001
AMYL	36.03 ab	9.00 b	28.62 ab	43.73 ab	19.18 b	49.28 a	0.001
BUN	10.73	10.57	10.92	13.43	12.23	10.96	0.11
Ca	2.22	2.21	2.25	2.19	2.25	2.25	0.21
CREA	100.18 a	104.67 a	81.13 b	76.42 bc	70.54 c	51.89d	0.001
CHOL	6.88 a	6.77 ab	6.68 ab	6.87 ab	6.18 bc	5.82 c	0.001
GGT	0.013	0.011	0.012	0.017	0.014	0.014	0.15
GLO	32.26 a	32.70 a	32.15 ab	32.20 ab	30.08 b	27.30 c	0.001
GLU	3.58 a	4.67 b	4.60 b	4.72 b	4.26 ab	4.42 ab	0.007
PHOS	2.07	2.04	1.92	2.02	2.06	1.91	0.07
TBL	4.51 ab	5.79 a	4.91 a	4.33 ab	3.51 b	2.85 c	0.001
TP	59.41 a	60.19 a	60.53 a	60.58 a	58.33 a	55.40 b	0.001
A/G	0.83 a	0.83 a	0.87 ab	0.86 ab	0.92 b	1.03 c	0.001

^§^ Means followed by different letters in rows are different at *p* < 0.05.

**Table 7 animals-12-02725-t007:** Least-squares means ± standard errors of the original serum analytes by inoculation dates ^§^.

Analyte	7–18 Oct. 2010	13 Dec. 2010	13–20 Sept. 2011	13 Dec. 2011	11–20 Sept. 2012	10–17 Sept. 2013	Prob
ALB	26.7 ± 0.3 a	27.3 ± 0.4 ab	28.1 ± 0.3 b	27.7 ± 0.6 ab	28.0 ± 0.3 b	28.2 ± 0.3 b	0.001
ALKP	76.4 ± 5.1	71.9 ± 7.1	68.8 ± 5.0	80.9 ± 9.5	74.4 ± 4.8	76.9 ± 4.6	0.68
ALT	80.1 ± 12.7 a	88.1 ± 20.7 ab	113.0 ± 10.7 ab	95.7 ± 20.5 ab	111.3 ± 10.2 ab	141.1 ± 9.9 b	0.001
AMYL	70.3 ± 8.5 ab	35.6 ± 13.9 b	61.7 ± 7.2 ab	87.0 ± 13.7 a	50.3 ± 6.8 ab	68.3 ± 6.5 ab	0.009
BUN	12.49 ± 0.85	11.90 ± 1.19	12.45 ± 0.83	14.66 ± 1.59	13.36 ± 0.79	11.72 ± 0.76	0.33
Ca	2.22 ± 0.02	2.21 ± 0.03	2.25 ± 0.02	2.19 ± 0.03	2.34 ± 0.01	2.25 ± 0.01	0.21
CREA	105.6 ± 3.9 a	106.1 ± 5.4 a	86.0 ± 3.8 b	81.8 ± 7.2 b	76.4 ± 3.6 b	52.4 ± 4.5c	0.001
CHOL	7.05 ± 0.16 a	6.92 ± 0.27 ab	6.83 ± 0.14 ab	6.98 ± 0.26 ab	6.27 ± 0.14 bc	5.89 ± 0.12c	0.001
GGT	4.22 ± 1.41	2.19 ± 2.28	3.10 ± 1.18	2.97 ± 2.25	0.78 ± 1.13	0.74 ± 1.06	0.14
GLO	33.9 ± 0.7 ab	34.0 ± 1.0 ab	34.3 ± 0.7 a	33.8 ± 1.4 ab	31.5 ± 0.7 b	27.6 ± 0.6c	0.001
GLU	3.69 ± 0.26 a	4.65 ± 0.43 ab	4.76 ± 0.22 b	4.82 ± 0.42 b	4.45 ± 0.21 ab	4.58 ± 0.20 ab	0.006
PHOS	2.65 ± 0.10	2.19 ± 0.16	2.06 ± 0.08	2.19 ± 0.16	2.21 ± 0.08	2.01 ± 0.07	0.08
TBL	5.34 ± 0.35 a	5.86 ± 0.57 a	5.83 ± 0.30 a	5.28 ± 0.57 a	4.33 ± 0.28 a	3.32 ± 0.27 b	0.001
TP	60.7 ± 0.8 ab	61.3 ± 1.1 ab	62.4 ± 0.8 a	61.8 ± 1.5 a	59.5 ± 0.7 ab	55.8 ± 0.7 c	0.001
A/G	0.81 ± 0.02 a	0.83 ± 0.03 a	0.85 ± 0.02 ab	0.85 ± 0.03 ab	0.91 ± 0.02 b	1.03 ± 0.02 c	0.001

^§^ Means followed by different letters in rows are different at *p* < 0.05.

**Table 8 animals-12-02725-t008:** Regression coefficients, standard errors, and odds ratios of the transformed and original data of serum analytes on the number of months post-inoculation.

	Transformed Data			Original Data		
Analyte	β ± SE	Prob	Odds Ratio	β ± SE	Prob	Odds Ratio
ALB	−0.04297 ± 3.59571	0.22	0.958	−0.0162 ± 0.0112	0.15	0.984
ALKP	−0.000641 ± 0.000660	0.33	0.999	−0.1826 ± 0.1797	0.31	0.833
ALT	−0.003089 ± 0.000795	0.001	0.997	−0.1404 ± 0.4054	0.005	0.869
AMYL	0.051304 ± 0.021193	0.016	1.053	0.0219 ± 0.2714	0.94	1.022
BUN	0.000301 ± 0.000911	0.75	1.000	−0.0335 ± 0.0300	0.26	0.967
Ca	−0.003252 ± 0.000876	0.001	0.997	−0.0022 ± 0.0006	0.001	0.998
CREA	−0.006722 ± 0.000687	0.001	0.993	−0.2224 ± 0.1368	0.001	0.801
CHOL	0.000072 ± 0.000304	0.81	1.000	0.0010 ± 0.0052	0.85	1.001
GGT	0.129875 ± 0.086693	0.13	1.138	−0.0334 ± 0.0447	0.46	0.967
GLO	−0.000003 ± 0.00000	0.001	0.999	−0.2392 ± 0.0261	0.001	0.787
GLU	−0.018087 ± 0.004023	0.001	0.982	−0.0330 ± 0.0083	0.001	0.967
PHOS	0.000355 ± 0.000585	0.54	1.000	−0.0011 ± 0.0031	0.74	0.999
TBL	−0.018401 ± 0.001649	0.001	0.982	−0.1087 ± 0.0112	0.001	0.897
TP	−0.0000001 ± 0.0000	0.001	0.999	−0.2559 ± 0.0280	0.001	0.774
A/G	0.007201 ± 0.001027	0.001	1.007	0.0098 ± 0.0006	0.001	1.010

**Table 9 animals-12-02725-t009:** Pearson correlation coefficients of original (above diagonal) and Box–Cox transformed data (below diagonal) among serum analytes and with log2 antibody titer at termination.

	TP	ALB	GLO	ALKP	ALT	GGT	BUN	CREA	CHOL	GLU	Ca	PHOS	AMYL	TBL	A/G	Titer
TP	.	0.23 **	0.94 **	0.39 **	0.13 **	0.49 **	0.36 **	0.50 **	0.32 **	0.10 *	0.12 *	0.30 **	0.31 **	0.55 **	−0.67 **	0.49 **
ALB	0.33 **	.	−0.10 *	0.01	0.06	−0.05	−0.19 **	−0.16 **	−0.12 **	0.01	0.47 **	−0.23 **	0.05	0.09	0.53 **	−0.18 **
GLO	0.87 **	−0.11 *	.	0.40 **	0.11 *	0.52 **	0.44 **	0.57 **	0.37 **	0.11 *	−0.04	0.38 **	0.29 **	0.54 **	-0.87 **	0.58 **
ALKP	0.34 *	0.18 **	0.23 **	.	0.31 **	0.63 **	0.38 **	0.23 **	0.32 **	0.10 *	−0.10 *	0.35 **	0.31 **	0.27 **	−0.28 **	0.13 **
ALT	0.21 **	0.12 *	0.16 **	0.35 **	.	0.17 **	0.07	0.04	−0.05	0.20 **	0.05	0.11 *	0.12 *	0.14 **	−0.06	0.15 **
GGT	0.16 **	0.03	0.12 **	0.35 **	0.18 **	.	0.37 **	0.37 **	0.32 **	0.05	−0.05	0.27 **	0.33 **	0.29 **	−0.38 **	0.17 **
BUN	0.20 **	−0.16 **	0.22 **	0.34 **	0.06	0.30 **	.	0.59 **	0.33 **	0.04	−0.34 **	0.70 **	0.46 **	0.17 **	−0.38 **	0.15 **
CREA	0.46 **	−0.15 **	0.53 **	0.11 *	0.06	0.14 **	0.47 **	.	0.37 **	−0.03	−0.12 *	0.50 **	0.27 **	0.46 **	−0.54 **	0.36 **
CHOL	0.29 **	−0.10 *	0.31 **	0.22 **	−0.07	0.11 *	0.34 **	0.33 **	.	0.01	−0.01	0.19 **	0.28 **	0.25 **	−0.34 **	0.21 **
GLU	0.11 *	0.02	0.11 *	0.13 **	0.19 **	0.07	−0.04	−0.03	0.02	.	−0.16 **	0.01	−0.01	0.24 **	−0.09	0.03
Ca	0.21 **	0.45 **	0.02	−0.06	0.07	−0.09	−0.28 **	−0.08	0.01	−0.16 **	.	−0.33 **	−0.14 **	0.01	0.20 **	−0.01
PHOS	0.14 **	−0.19 **	0.22 **	0.25 **	0.11 *	0.18 **	0.51 **	0.42 **	0.15 **	−0.15 **	−0.27 **	.	0.35 **	0.32 **	−0.37 **	0.15 **
AMYL	0.08	0.01	0.01	0.24 **	0.02	0.27 **	0.42 **	0.08	0.26 **	−0.06	−0.08	0.18 **	.	0.12 **	−0.16 **	0.21 **
TBL	0.55 **	0.11 *	0.54 **	0.13 **	0.20 **	0.04	−0.03	0.44 **	0.19 **	0.23 **	0.05	0.16 **	0.14 **	.	−0.42 **	0.26 **
A/G	−0.55 **	0.55 **	−0.88 **	−0.10 *	−0.08	−0.09	0.25 **	−0.50 **	−0.29 **	−0.08	0.19 **	−0.28 **	−0.01	0.41 **	.	−0.59 **
Titer	0.53 **	−0.17 **	0.62 **	0.14 **	0.16 **	0.16 **	0.25 **	0.36 **	0.20 **	0.04	−0.04	0.20 **	0.20	0.26 **	−0.60 **	.

Differences between correlation coefficients were significant at *p* < 0.05 (*) or *p* < 0.01 (**).

## Data Availability

Data supporting reported results are contained within the article. All datasets collected and analyzed during the current study are available from the corresponding author on reasonable request.

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
