# Peer review of "Serum Analytes of American Mink (Neovison Vison) Challenged with Aleutian Mink Disease Virus"

_animals, 2022, doi:10.3390/ani12202725_

Round 1

Reviewer 1 Report

This is a detailed manuscript examining the effects of previous election for tolerance on serum profiles of mink challenged with AMDV (Aleutian mink disease virus). While the manuscript is well-written, there is too much information in the results section and the discussion section is too long. That can detract from the main messages. Suggest moving some information to supplementary data. In the discussion, please avoid repetition of the results and only discuss how that the results relate to literature and future implications.

For tables, please add a footnote indicating what does a,b,c stands for.

Line 273: AMYL instead of AMY?

Author Response

Q1-The reviewer mentioned that “there is too much information in the results section and the discussion section is too long.  That can detract from the main messages. Suggest moving some information to supplementary data”.

A: During writing this paper, we were concerned about the large body of data that we thought necessary to report and we were worried about the large size of the paper. We agree that the paper is long. We, however, cannot decide which part to delete or move, We would be glad to move some information to supplementary data if the reviewer can make a suggestion on which parts should be moved or even deleted.   Table 10 was moved to Supplementary data.

Q2: The reviewer suggested that “In the discussion, please avoid repetition of the results and only discuss how that the results relate to literature and future implications.”

A; Results of our study were removed in 20 cases (lines 423, 439, 461, 467, 485, 503, 513, 527, 547, 553, 560, 670, 681, 693, 698, 700, 701, 708, 709, 724).

Q3: For tables, please add a footnote indicating what does a,b,c stands for.

A:The footnote of Table 3 was  “In this and other tables, means followed by different superscript letters in rows are different at P<0.05”.

The footnotes of all tables were changed to “Means followed by different superscript letters in rows are different at P<0.05”.

Line 273: AMYL instead of AMY? Corrected

Reviewer 2 Report

The manuscript titled “Serum Analytes of American Mink (Neovison vison) Challenged 2 with Aleutian Mink Disease Virus” is so interesting for this journal and this present important results about Aleutian Mink disease Virus. The manuscript is well presented and well written, however some minor changes should be done before ecptance.

1.      The tables should be revised. Numbers and letter are not aligned and should be revise all of them.

2.      In my opinion there are too many information in form of tables and it is hard for the reader follow the manuscript with so big tables. Could the authors think about submit as supplementary material some of these tables (as the case of table 10)?

3.      Table 10 with the references should be rethinked because it cannot be all this information with the refences below.

4.      The discussion should be shortened. It is so difficult to follow for the reader when the discussion is longer that the rest of the paper.

Author Response

  1. The tables should be revised. Numbers and letter are not aligned and should be revise all of them
  2. A: We tried not to use Table format, which resulted in little alignment of numbers and letters. We thought this was the journals policy. If needed, were revise all tables using Table formats.

Q2. In my opinion there are too many information in form of tables and it is hard for the reader follow the manuscript with so big tables. Could the authors think about submit as supplementary material some of these tables (as the case of table 10)?

A: The same concern as Q1 of the reviewer 1. We need the reviewer’s suggestions as which part should be moved. Table 10 was moved to Supplementary data.

   Q3.  Table 10 with the references should be rethinked because it cannot be all this information with the refences below.

A: Table 10 was moved to Supplementary data.

Q4.     The discussion should be shortened. It is so difficult to follow for the reader when the discussion is longer that the rest of the paper.

A: We are aware of the long discussions. This in fact is a mini-review. We spent almost 10 months reading all available papers in detail and incorporate them into the discussion.  To the best of our knowledge, most of the discussions have not been reported before. We would greatly appreciate it if the reviewer agreed in keeping the Discussion.

Reviewer 3 Report

The study is relevant, innovative, and well described. The effects of AMDV infection on blood analytes of mink was restricted to only one report.

1.      What do you mean by black American mink (Neovison vison). Species name is American mink?

2.      Title, body, and footnotes of the table must be organised in one page.

3.      There is contradiction between simple summary “Total serum protein and globulin were found to be the most useful biomarkers of tolerance, whereas the relationships of other serum analytes to tolerance were weak or negligible” and abstract “Significant differences were observed among tolerant groups in the concentrations of globulin (GLO), total protein (TP), alkaline phosphatase, urea nitrogen and calcium.”

4.      Please cite the following article Zaleska-Wawro et al. 2021 doi: 10.3390/ani11102975. 

5.      L146 conditions on the counter-immunoelectrophoresis is needed.

6.      2.1-2.3 ethical statements on the use of animals should be included also in methodical part of the manuscript.

7.      Tables 9 and 10 are not tables they are figures. Please fix it.

Author Response

.      What do you mean by black American mink (Neovison vison). Species name is American mink?

A: We agree. Neovison vison is American mink. We used the word “black” to specify the colour type that we used. There are other “American mink” with different colour phases which may have different serum analytes.

Q2.      Title, body, and footnotes of the table must be organised in one page.

A: This is followed, except for Table 10 which was moved to Supplementary data.

.

Q3.      There is contradiction between simple summary “Total serum protein and globulin were found to be the most useful biomarkers of tolerance, whereas the relationships of other serum analytes to tolerance were weak or negligible” and abstract “Significant differences were observed among tolerant groups in the concentrations of globulin (GLO), total protein (TP), alkaline phosphatase, urea nitrogen and calcium.”

A; We assumed that inoculated mink with low antibody titers were able to tolerate the infection, and because TP and GLO had the greatest correlation coefficients with antibody titer, we concluded that (Page 658)  “the finding that the greatest correlation coefficients between antibody titer and serum analytes were for TP (0.53), GLO (0.62) and A/G (-.60) (Table 9), suggested their strong associations with the degree of tolerance of mink to AMDV infection.”   

Although significant differences were observed among tolerant groups in the concentrations of globulin (GLO), total protein (TP), alkaline phosphatase, urea nitrogen and calcium (as stated in the Abstract), the rankings of GLO and TP were parallel with the percentage of tolerant ancestry. Therefore, the two statements are correct, and if the reviewer suggests, we can reword the sentences.

Q4.      Please cite the following article Zaleska-Wawro et al. 2021 doi: 10.3390/ani11102975. 

A: We thoroughly read this review article, but did not find any information over and above what we already have in our paper. We would appreciate it very much if the reviewer could please help us in finding a relevant sentence in that review that could be inserted into our paper.

Q5.      L146 conditions on the counter-immunoelectrophoresis is needed.

A: The following sentence was added after line 146. “The standard CIEP test was performed on plasma at the Animal Health Laboratory of the Nova Scotia Department of Agriculture in Truro, Nova Scotia, Canada, which is accredited for this test by the Standards Council of Canada. The test was performed using a cell-cultured antigen supplied by the United Vaccine, Inc., Madison, Wisconsin, USA”.

Q6.      2.1-2.3 ethical statements on the use of animals should be included also in methodical part of the manuscript.

A: The following sentence was added to section 2-1” Standard Operating Procedures were prepared for animal management and sampling according to the standards of the Canadian Council for Animal Care.

Q7.      Tables 9 and 10 are not tables they are figures. Please fix it.

A: Unfortunately, the suggestion of the reviewer is not clear to us. Please elaborate on the changes that we need to make.